# Estimation of Skill Distribution from a Tournament

**Ali Jadbabaie**
Department of CEE
Massachusetts Institute of Technology
Cambridge, MA 02139
jadbabai@mit.edu

**Anuran Makur**
Department of EECS
Massachusetts Institute of Technology
Cambridge, MA 02139
a_makur@mit.edu

**Devavrat Shah**
Department of EECS
Massachusetts Institute of Technology
Cambridge, MA 02139
devavrat@mit.edu

## Abstract

In this paper, we study the problem of learning the skill distribution of a population of agents from observations of pairwise games in a tournament. These games are played among randomly drawn agents from the population. The agents in our model can be individuals, sports teams, or Wall Street fund managers. Formally, we postulate that the likelihoods of outcomes of games are governed by the parametric Bradley-Terry-Luce (or multinomial logit) model, where the probability of an agent beating another is the ratio between its skill level and the pairwise sum of skill levels, and the skill parameters are drawn from an unknown, non-parametric skill density of interest. The problem is, in essence, to learn a distribution from noisy, quantized observations. We propose a surprisingly simple and tractable algorithm that learns the skill density with near-optimal minimax mean squared error scaling as $n^{-1+\varepsilon}$, for any $\varepsilon > 0$, so long as the density is smooth. Our approach brings together prior work on learning skill parameters from pairwise comparisons with kernel density estimation from non-parametric statistics. Furthermore, we prove information theoretic lower bounds which establish minimax optimality of the skill parameter estimation technique used in our algorithm. These bounds utilize a continuum version of Fano's method along with a careful covering argument. We apply our algorithm to various soccer leagues and world cups, cricket world cups, and mutual funds. We find that the entropy of a learnt distribution provides a quantitative measure of skill, which in turn provides rigorous explanations for popular beliefs about perceived qualities of sporting events, e.g., soccer league rankings. Finally, we apply our method to assess the skill distributions of mutual funds. Our results shed light on the abundance of low quality funds prior to the Great Recession of 2008, and the domination of the industry by more skilled funds after the financial crisis.

## 1 Introduction

It is a widely-held belief among soccer enthusiasts that English Premier League (EPL) is the most competitive amongst professional leagues even though the likely eventual winner is often one of a handful of usual suspects [1, 2]. Similarly, the Cricket World Cup in 2019 is believed to be the most exciting in the modern history of the sport, and ended with one of the greatest matches of all

---

The author ordering is alphabetical.

Table 1: Comparison of our contributions with prior works. The notation $\tilde{O}$ and $\tilde{\Omega}$ hide $\mathrm{poly}(\log(n))$ terms, and $\varepsilon > 0$ is any arbitrarily small constant.

| Estimation problem | Loss function | Upper bound | Lower bound |
|---|---|---|---|
| Smooth $\mathcal{C}^\infty$ skill PDF | MSE | $\tilde{O}(n^{-1+\varepsilon})$ (Theorem 3) | $\Omega(n^{-1})$ [5, 9] |
| BTL skill parameters | relative $\ell^\infty$-norm | $\tilde{O}(n^{-1/2})$ [10] | $\tilde{\Omega}(n^{-1/2})$ (Theorem 1) |
| BTL skill parameters | $\ell^1$-norm | $O(n^{-1/2})$ [10] | $\tilde{\Omega}(n^{-1/2})$ (Theorem 2) |

time [3, 4]. But is any of this backed up by data, or are they just common misconceptions? In this work, we answer this question by quantifying such observations, beyond mere sports punditry and subjective opinions, in a data-driven manner. We then illustrate that a similar approach can be used to quantify the evolution of the overall quality and relative skills of mutual funds over the years.

To this end, we posit that the population of agents in a tournament, e.g., EPL teams or mutual fund managers, has an associated distribution of skills with a probability density function (PDF) $P_\alpha$ over $\mathbb{R}_+$. Our goal is to learn this $P_\alpha$. Traditionally, in the non-parametric statistics literature, cf. [5], one observes samples from the distribution directly to estimate $P_\alpha$. In our setting, however, we can only observe extremely noisy, quantized values. Specifically, given $n$ individuals, teams, or players participating in a tournament, indexed by $[n] \triangleq \{1, \ldots, n\}$, let their skill levels be $\alpha_i, i \in [n]$, which are sampled independently from $P_\alpha$. We observe the outcomes of pairwise games or comparisons between them. More precisely, for each $i \neq j \in [n]$, with probability $p \in (0, 1]$, we observe the outcomes of $k \geq 1$ games, and with probability $1 - p$, we observe nothing. Let $\mathcal{G}(n, p)$ denote the induced Erdős-Rényi random graph on $[n]$ with edge $\{i, j\} \in \mathcal{G}(n, p)$ if games between $i$ and $j$ are observed. (Note that $\mathcal{G}(n, p)$ is independent of $\alpha_1, \ldots, \alpha_n$.) For $\{i, j\} \in \mathcal{G}(n, p)$, let $Z_m(i, j) \in \{0, 1\}$ denote whether $j$ beats $i$, i.e., value 1 if $j$ beats $i$ and 0 otherwise, in game $m \in [k]$. By definition, $Z_m(i, j) + Z_m(j, i) = 1$. We assume the *Bradley-Terry-Luce (BTL)* [6, 7] or *multinomial logit model* [8] where:

$$\mathbb{P}(Z_m(i, j) = 1 \mid \alpha_1, \ldots, \alpha_n) \triangleq \frac{\alpha_j}{\alpha_i + \alpha_j}, \tag{1}$$

independently of the outcomes of all other games. Our objective is to learn $P_\alpha$ from the observations $\{Z_m(i, j) : \{i, j\} \in \mathcal{G}(n, p), m \in [k]\}$, instead of $\alpha_i, i \in [n]$ (as in traditional statistics [5]). For a given, fixed set of $\alpha_i, i \in [n]$, learning them from pairwise comparison data $\{Z_m(i, j) : \{i, j\} \in \mathcal{G}(n, p), m \in [k]\}$ has been extensively studied in the recent literature [10–12]. Nevertheless, this line of research does not provide any means to estimate the underlying skill distribution $P_\alpha$.

**Contributions.** As the main contribution of this work, we develop a statistically near-optimal and computationally tractable method for estimating the skill distribution $P_\alpha$ from a subset of pairwise comparisons. Our estimation method is a two-stage algorithm that uses the (spectral) rank centrality estimator [11, 12] followed by the Parzen-Rosenblatt kernel density estimator [13, 14] with carefully chosen bandwidth. We establish that the minimax *mean squared error (MSE)* of our method scales as $\tilde{O}(n^{-\eta/(\eta+1)})$ for any $P_\alpha$ belonging to an $\eta$-Hölder class. Thus, if $P_\alpha$ is smooth ($\mathcal{C}^\infty$) with bounded derivatives, then the minimax MSE is $\tilde{O}(n^{-1+\varepsilon})$ for any $\varepsilon > 0$; see Theorem 3 for details. Somewhat surprisingly, although we do not directly observe $\alpha_i, i \in [n]$, this minimax MSE rate matches the minimax MSE lower bound of $\Omega(n^{-1})$ for smooth $P_\alpha$ even when $\alpha_i, i \in [n]$ are observed [5, 9].

As a key step in our estimation method, we utilize the rank centrality algorithm [11, 12] for estimating $\alpha_i, i \in [n]$. While the optimal learning rate of the rank centrality algorithm with respect to relative $\ell^2$-loss is well-understood [10–12], the optimal learning rates with respect to relative $\ell^\infty$ and $\ell^1$-losses are not known since we only know upper bounds [10], but not matching minimax lower bounds. In Theorems 1 and 2, we prove minimax lower bounds of $\tilde{\Omega}(n^{-1/2})$ with respect to both relative $\ell^\infty$ and $\ell^1$-losses. These bounds match the learning rates of the rank centrality algorithm obtained in [10] with respect to both $\ell^\infty$ and $\ell^1$-losses, and hence, identify the optimal minimax rates. We derive these information theoretic lower bounds by employing a recent variant of the generalized Fano's method with covering arguments. (Our main technical results are all delineated in Table 1.)

Finally, we illustrate the utility of our algorithm through four experiments on real-world data: cricket world cups, soccer world cups, European soccer leagues, and mutual funds. Intuitively, a concentrated

skill distribution, i.e., one that is close to a Dirac delta measure, corresponds to a balanced tournament with players that are all equally skilled. Hence, the outcomes of games are random or unpredictable. On the other hand, a skill distribution that is close to uniform suggests a wider spread of players' skill levels. So, the outcomes of games are driven more by skill rather than luck (or random chance). We, therefore, propose to use the negative entropy of a learnt skill distribution as a way to measure the "overall skill score," because negative entropy captures distance to the uniform distribution. For cricket world cups, we find that negative entropy decreases from 2003 to 2019. Indeed, this corroborates with fan experience, where in 2003, Australia and India dominated but all other teams were roughly equal, while in 2019, there was a healthy spread of skill levels making many teams potential contenders for the championship. In soccer, we observe that the EPL and World Cup have high negative entropy, which indicates that most teams are competitive, and thus, it is very difficult to predict outcomes up front. Lastly, the negative entropy of US mutual funds decreases significantly during the Great Recession of 2008, and we see flatter skill distributions post 2008. This reveals that mutual funds became more competent to avoid being weeded out of the market by the financial crisis.

It is worth mentioning that there are several reasons to estimate $P_\alpha$ rather than the individual skill levels $\alpha_1, \ldots, \alpha_n$. Although specific functionals of $P_\alpha$, e.g., moments or variance, may be directly estimated from estimates of skill levels, estimating $P_\alpha$ simultaneously recovers information about all such functionals. Indeed, it can be shown that MSE guarantees for estimating $P_\alpha$ yield uniform guarantees on estimating bounded statistics of the form $\mathbb{E}[f(\alpha)]$ for functions $f : \mathbb{R}_+ \to \mathbb{R}$. Since different functionals are pertinent for different applications, a good estimate of the skill distribution $P_\alpha$ is very useful. For example, we utilize negative entropy of $P_\alpha$ to define overall skill scores. Standard non-parametric plug-in estimators for entropy in the literature, e.g., integral, resubstitution, or splitting data estimators [15], require an estimate of $P_\alpha$ to compute entropy. Therefore, in the context of this work, estimating $P_\alpha$ is eminently desirable.

**Related work.** A long line of related work [5–14, 16–33] pertaining to measuring the overall skill levels in tournaments, non-parametric density estimation, the BTL and related models, classical algorithms to estimate BTL model parameters, and modern non-asymptotic analyses of such algorithms are presented and discussed in [34, Section 1] due to space constraints.

**Notational preliminaries.** We briefly introduce some relevant notation. Let $\mathbb{N} \triangleq \{1, 2, 3, \ldots\}$ denote the set of natural numbers. For any $n \in \mathbb{N}$, let $\mathcal{S}_n$ denote the probability simplex of row probability vectors in $\mathbb{R}^n$, and $\mathcal{S}_{n \times n}$ denote the set of all $n \times n$ row stochastic matrices in $\mathbb{R}^{n \times n}$. For any vector $x \in \mathbb{R}^n$ and any $q \in [1, \infty]$, let $\|x\|_q$ denote the $\ell^q$-norm of $x$. Moreover, $\log(\cdot)$ denotes the natural logarithm function with base $e$, $\mathbb{1}\{\cdot\}$ denotes the indicator function that equals 1 if its input proposition is true and 0 otherwise, and $\lceil \cdot \rceil$ denotes the ceiling function. Finally, we will use standard Bachmann-Landau asymptotic notation, e.g., $O(\cdot), \Omega(\cdot), \Theta(\cdot)$, where it is understood that $n \to \infty$, and tilde notation, e.g., $\tilde{O}(\cdot), \tilde{\Omega}(\cdot), \tilde{\Theta}(\cdot)$, when we neglect $\mathsf{poly}(\log(n))$ factors and problem parameters other than $n$.

## 2 Estimation algorithm

**Overview.** Our interest is in estimating the skill PDF $P_\alpha$ from noisy, discrete observations $\{Z_m(i,j) : \{i,j\} \in \mathcal{G}(n,p), m \in [k]\}$. Instead, if we had exact knowledge of the samples $\alpha_i, i \in [n]$ from $P_\alpha$, then we could utilize traditional methods from non-parametric statistics such as kernel density estimation. However, we do not have access to these samples. So, given pairwise comparisons $\{Z_m(i,j) : \{i,j\} \in \mathcal{G}(n,p), m \in [k]\}$ generated as per the BTL model with parameters $\alpha_i, i \in [n]$, we can use some recent developments from the BTL-related literature to estimate these skill parameters first. Therefore, a natural two-stage algorithm is to first estimate $\alpha_i, i \in [n]$ using the observations, and then use these estimated parameters to produce an estimate of $P_\alpha$. We do precisely this. The key challenge is to ensure that the PDF estimation method is robust to the estimation error in $\alpha_i, i \in [n]$. As our main contribution, we rigorously argue that carefully chosen methods for both steps produces as good an estimation of $P_\alpha$ as if we had access to the exact knowledge of $\alpha_i, i \in [n]$.

**Setup.** We formalize the setup here. For any given $\delta, \epsilon, b \in (0, 1)$ and $\eta, L_1, B > 0$, let $\mathcal{P} = \mathcal{P}(\delta, \epsilon, b, \eta, L_1, B)$ be the set of all uniformly bounded PDFs with respect to the Lebesgue measure on $\mathbb{R}$ that have support in $[\delta, 1]$, belong to the $\eta$-*Hölder class* [5, Definition 1.2], and are lower bounded by $b$ in an $\epsilon$-neighborhood of 1. More precisely, for every $f \in \mathcal{P}$, $f$ is bounded (almost everywhere), i.e., $f(x) \leq B$ for all $x \in [\delta, 1]$; $f$ is $s = \lceil \eta \rceil - 1$ times differentiable, and its $s$th

derivative $f^{(s)} : [\delta, 1] \to \mathbb{R}$ satisfies $|f^{(s)}(x) - f^{(s)}(y)| \leq L_1 |x - y|^{\eta - s}$ for all $x, y \in [\delta, 1]$; and $f(x) \geq b$ for all $x \in [1 - \epsilon, 1]$. As an example, when $\eta = 1$, $\mathcal{P}$ denotes the set of all Lipschitz continuous PDFs on $[\delta, 1]$ that are lower bounded near 1. Furthermore, we define the observation matrix $Z \in [0, 1]^{n \times n}$, whose $(i, j)$th entry is:

$$\forall i, j \in [n], \; Z(i, j) \triangleq \begin{cases} \mathbb{1}\{\{i, j\} \in \mathcal{G}(n, p)\} \frac{1}{k} \sum_{m=1}^{k} Z_m(i, j), & i \neq j, \\ 0, & i = j. \end{cases} \tag{2}$$

**Estimation error.** It turns out that $Z$ is a sufficient statistic for the purposes of estimating $\alpha_i, i \in n$ [10, p.2208]. For this reason, we shall restrict our attention to all possible estimators of $P_\alpha$ using $Z$. Specifically, let $\widehat{\mathcal{P}}$ be set of all possible measurable and potentially randomized estimators that map $Z$ to a Borel measurable function from $\mathbb{R}$ to $\mathbb{R}$. Then, the minimax MSE risk is defined as:

$$R_{\mathsf{MSE}}(n) \triangleq \inf_{\hat{P} \in \widehat{\mathcal{P}}} \sup_{P_\alpha \in \mathcal{P}} \mathbb{E}\left[ \int_{\mathbb{R}} \left( \hat{P}(x) - P_\alpha(x) \right)^2 \mathrm{d}x \right] \tag{3}$$

where the expectation is with respect to the randomness in $Z$ as well as within the estimator. Our interest will be in understanding the scaling of $R_{\mathsf{MSE}}(n)$ as a function of $n$ and $\eta$. In the sequel, we will assume that the parameters $k, p, \delta, \epsilon, b$ can depend on $n$, and all other parameters are constant.

**Step 1: Estimate $\alpha_i, i \in [n]$.** Given the observation matrix $Z$, let $S \in \mathbb{R}^{n \times n}$ be the "empirical stochastic matrix" whose $(i, j)$th element is given by:

$$\forall i, j \in [n], \; S(i, j) \triangleq \begin{cases} \dfrac{1}{2np} Z(i, j), & i \neq j, \\ 1 - \dfrac{1}{2np} \sum_{r=1}^{n} Z(i, r), & i = j. \end{cases} \tag{4}$$

As shown in [34, Proposition 3], it is straightforward to verify that $S \in \mathcal{S}_{n \times n}$ (i.e., $S$ is row stochastic) with high probability when $p = \Omega(\log(n)/n)$. Next, inspired by the *rank centrality* algorithm in [11, 12], let $\hat{\pi}_* \in \mathcal{S}_n$ be the invariant distribution of $S$, given by:

$$\hat{\pi}_* \triangleq \begin{cases} \text{invariant distribution of } S \text{ such that } \hat{\pi}_* = \hat{\pi}_* S, & S \in \mathcal{S}_{n \times n}, \\ \text{any randomly chosen distribution in } \mathcal{S}_n, & S \notin \mathcal{S}_{n \times n}, \end{cases} \tag{5}$$

where when $S \in \mathcal{S}_{n \times n}$, an invariant distribution always exists and we choose one arbitrarily when it is not unique. Then, we can define the following estimates of $\alpha_1, \ldots, \alpha_n$ based on $Z$:

$$\forall i \in [n], \; \hat{\alpha}_i \triangleq \frac{\hat{\pi}_*(i)}{\|\hat{\pi}_*\|_\infty} \tag{6}$$

where $\hat{\pi}_*(i)$ denotes the $i$th entry of $\hat{\pi}_*$ for $i \in [n]$.

**Step 2: Estimate $P_\alpha$.** Using (6), we construct the *Parzen-Rosenblatt (PR) kernel density estimator* $\widehat{\mathcal{P}}^* : \mathbb{R} \to \mathbb{R}$ for $P_\alpha$ based on $\hat{\alpha}_1, \ldots, \hat{\alpha}_n$ (instead of $\alpha_1, \ldots, \alpha_n$) [13, 14]:

$$\forall x \in \mathbb{R}, \; \widehat{\mathcal{P}}^*(x) \triangleq \frac{1}{nh} \sum_{i=1}^{n} K\left( \frac{\hat{\alpha}_i - x}{h} \right) \tag{7}$$

where $h > 0$ is a judiciously chosen bandwidth parameter (see the proof in [34, Appendix B.2]):

$$h = \gamma \max\left\{ \frac{1}{\delta^{\frac{1}{\eta + 1}} (pk)^{\frac{1}{2\eta + 2}}}, 1 \right\} \left( \frac{\log(n)}{n} \right)^{\frac{1}{2\eta + 2}} \tag{8}$$

for any (universal) constant $\gamma > 0$, and $K : [-1, 1] \to \mathbb{R}$ is any fixed kernel function with certain properties that we explain below.

For any $s \in \mathbb{N} \cup \{0\}$, the function $K : [-1, 1] \to \mathbb{R}$ is said to be a *kernel of order $s$*, where we assume that $K(x) = 0$ for $|x| > 1$, if $K$ is (Lebesgue) square-integrable, $\int_{\mathbb{R}} K(x) \, \mathrm{d}x = 1$, and $\int_{\mathbb{R}} x^i K(x) \, \mathrm{d}x = 0$ for all $i \in [s]$ when $s \geq 1$. Such kernels of order $s$ can be constructed using orthogonal polynomials as expounded in [5, Section 1.2.2]. We will additionally assume that there exists a constant $L_2 > 0$ such that our kernel $K : [-1, 1] \to \mathbb{R}$ is *$L_2$-Lipschitz continuous*, i.e.,

$|K(x) - K(y)| \leq L_2|x - y|$ for all $x, y \in \mathbb{R}$. This is a mild assumption since several well-known kernels satisfy it. For instance, the (parabolic) *Epanechnikov kernel* $K_{\mathsf{E}}(x) \triangleq \frac{3}{4}(1 - x^2)\mathbb{1}\{|x| \leq 1\}$ has order $s = 1$, and is Lipschitz continuous with $L_2 = \frac{3}{2}$ [17]. Other examples of valid kernels can be found in [5, p.3 and Section 1.2.2].

**Algorithm, in summary.** Here, we provide the 'pseudo-code' summary of our algorithm.

---
**Algorithm 1** Estimating skill PDF $P_\alpha$ using $Z$.

---
**Input:** Observation matrix $Z \in [0, 1]^{n \times n}$ (as defined in (2))
**Output:** Estimator $\widehat{\mathcal{P}}^* : \mathbb{R} \to \mathbb{R}$ of the unknown PDF $P_\alpha$
    *Step 1:* **Skill parameter estimation using rank centrality algorithm**
1: Construct $S \in \mathcal{S}_{n \times n}$ according to (4) using $Z$ (and $p$ and $n$)
2: Compute leading left eigenvector $\hat{\pi}_* \in \mathcal{S}_n$ of $S$ in (5)         ▷ $\hat{\pi}_*$ is the invariant distribution of $S$
3: Compute estimates $\hat{\alpha}_i = \hat{\pi}_*(i)/\|\hat{\pi}_*\|_\infty$ for $i = 1, \ldots, n$ via (6)
    *Step 2:* **Kernel density estimation using Parzen-Rosenblatt method**
4: Compute bandwidth $h$ via (8) (using $p$, $k$, $\delta$, $\eta$, and $n$)
5: Construct $\widehat{\mathcal{P}}^*$ according to (7) using $\hat{\alpha}_1, \ldots, \hat{\alpha}_n$, $h$, and a valid kernel $K : [-1, 1] \to \mathbb{R}$
6: **return** $\widehat{\mathcal{P}}^*$

---

With fixed $\delta \in (0, 1)$, $\eta > 0$, and a valid kernel $K : [-1, 1] \to \mathbb{R}$, and given knowledge of $k \in \mathbb{N}$ and $p \in (0, 1]$ (which can also be easily estimated), Algorithm 1 constructs the estimator (7) for $P_\alpha$ based on $Z$. In Algorithm 1, we assume that $S \in \mathcal{S}_{n \times n}$, because this is almost always the case in practice. Furthermore, if $k$ varies between players so that $i$ and $j$ play $k_{i,j} = k_{j,i}$ games for $i \neq j$, we can re-define the data $Z(i, j)$ to use $k_{i,j}$ instead of $k$ in (2), and utilize an appropriately altered bandwidth $h$. For example, we can use $k' = \min_{\{i,j\} \in \mathcal{G}(n,p)} k_{i,j}$ in place of $k$ in (8) to define $h$, which would yield theoretical guarantees akin to Theorem 3 with $k'$. The computational complexity of Algorithm 1 is determined by the running time of rank centrality, e.g., if the spectral gap of $S$ is $\Theta(1)$ and we use *power iteration* (cf. [35, Section 7.3.1], [36, Section 4.4.1]) to obtain an $O(n^{-5})$ $\ell^2$-approximation of $\hat{\pi}_*$, then Algorithm 1 runs in $O(n^2 \log(n))$ time. We refer readers to [34, Appendix B.1] for further intuition regarding Algorithm 1.

## 3 Main results

We now present our main results: an achievable minimax MSE for the $P_\alpha$ estimation method in Algorithm 1, and minimax lower bounds on estimation of the skill parameters $\alpha_i, i \in [n]$ from $Z$ (i.e., Step 1 of Algorithm 1) for *any* method. This collectively establishes the near-optimality of our proposed method as $\eta \to \infty$, i.e., as the density becomes smooth ($\mathcal{C}^\infty$). To this end, we first establish minimax rates for skill parameter estimation, and then derive minimax rates for PDF estimation.

**Tight minimax bounds on skill parameter estimation.** To obtain tight $P_\alpha$ estimation, it is essential that we have tight skill parameter estimation. Hence, we show that the parameter estimation step performed in (5) has minimax optimal rate. Specifically, we define the "canonically scaled" skill parameters $\pi \in \mathcal{S}_n$ with $i$th entry given by:

$$\forall i \in [n], \ \ \pi(i) \triangleq \frac{\alpha_i}{\alpha_1 + \cdots + \alpha_n} \,. \tag{9}$$

Building upon [10, Theorem 3.1], the ensuing theorem portrays that the *minimax relative $\ell^\infty$-risk* of estimating (9) based on $Z$ is $\tilde{\Theta}(n^{-1/2})$ (see Table 1). For simplicity, we will assume throughout this subsection on skill parameter estimation that $\delta$, $p$, and $k$ are $\Theta(1)$.

**Theorem 1** (Minimax Relative $\ell^\infty$-Risk)**.** *For sufficiently large constants $c_{14}, c_{15} > 0$ (which depend on $\delta$, $p$, and $k$), and for all sufficiently large $n \in \mathbb{N}$:*

$$\frac{c_{14}}{\log(n)\sqrt{n}} \leq \inf_{\hat{\pi}} \sup_{P_\alpha \in \mathcal{P}} \mathbb{E}\left[\frac{\|\hat{\pi} - \pi\|_\infty}{\|\pi\|_\infty}\right] \leq \sup_{P_\alpha \in \mathcal{P}} \mathbb{E}\left[\frac{\|\hat{\pi}_* - \pi\|_\infty}{\|\pi\|_\infty}\right] \leq c_{15}\sqrt{\frac{\log(n)}{n}}$$

*where the infimum is over all estimators $\hat{\pi} \in \mathcal{S}_n$ of $\pi$ based on $Z$, and $\hat{\pi}_* \in \mathcal{S}_n$ is defined in (5).*

The proof of Theorem 1 can be found in [34, Appendix C.3]. Theorem 1 states that the rank centrality estimator $\hat{\pi}_*$ achieves an extremal Bayes relative $\ell^\infty$-risk of $\tilde{O}(n^{-1/2})$, and no other estimator can achieve a risk that decays faster than $\tilde{\Omega}(n^{-1/2})$. In the same vein, we show that the *minimax (relative)* $\ell^1$-*risk* (or total variation distance risk) of estimating (9) based on $Z$ is also $\tilde{\Theta}(n^{-1/2})$ (see Table 1).

**Theorem 2** (Minimax $\ell^1$-Risk). *For sufficiently large constants $c_{17}, c_{18} > 0$ (which depend on $\delta$, $p$, and $k$), and for all sufficiently large $n \in \mathbb{N}$:*

$$\frac{c_{17}}{\log(n)\sqrt{n}} \leq \inf_{\hat{\pi}} \sup_{P_\alpha \in \mathcal{P}} \mathbb{E}[\|\hat{\pi} - \pi\|_1] \leq \sup_{P_\alpha \in \mathcal{P}} \mathbb{E}[\|\hat{\pi}_* - \pi\|_1] \leq \frac{c_{18}}{\sqrt{n}}\,.$$

Theorem 2 is established in [34, Appendix C.4]. The upper bounds in Theorems 1 and 2 follow from [10, Theorems 3.1 and 5.2] after some calculations, but the *lower bounds are novel contributions*. We prove them by first lower bounding the minimax risks in terms of Bayes risks in order to circumvent an involved analysis of the infinite-dimensional parameter space $\mathcal{P}$. In particular, we *set $P_\alpha \in \mathcal{P}$ to be the uniform PDF* over $[\delta, 1]$, denoted $\mathsf{unif}([\delta, 1]) \in \mathcal{P}$. We then lower bound the Bayes risks using a recent generalization of *Fano's method* [18, 19] (cf. [5, 37]), which was specifically developed to produce such lower bounds in the setting where the parameter space is a continuum, e.g., $[\delta, 1]$, instead of a finite set [38–41]; see [34, Appendices C.1 and C.2].

The principal analytical difficulty in executing the generalized Fano's method is in deriving a tight upper bound on the *mutual information* between $\pi$ and $Z$, denoted $I(\pi; Z)$ (see [42, Definition 2.3] for a formal definition), where the probability law of $\pi$ is defined using $P_\alpha = \mathsf{unif}([\delta, 1])$. The ensuing proposition presents our upper bound on $I(\pi, Z)$.

**Proposition 1** (Covering Number Bound on Mutual Information). *For all $n \geq 2$, we have:*

$$I(\pi; Z) \leq \frac{1}{2} n \log(n) + \frac{(1-\delta)^2}{8\delta^2}\left(2 + \delta + \frac{1}{\delta}\right) kpn\,.$$

Proposition 1 is proved in [34, Appendix A.2]. We note that although standard information inequalities, e.g. [40, Equation (44)], typically suffice to obtain minimax rates for various estimation problems, they only produce a sub-optimal estimate $I(\pi; Z) = O(n^2)$ in our problem, as explained at the end of [34, Appendix A.2]. So, to derive the sharper estimate $I(\pi; Z) = O(n \log(n))$ in Proposition 1, we execute a careful covering number argument that is inspired by the techniques of [43] (also see the distillation in [44, Lemma 16.1]).

We make two further remarks. Firstly, it is worth juxtaposing our results with [10, Theorem 5.2] and [12, Theorems 2 and 3], which state that the minimax relative $\ell^2$-risk of estimating $\pi$ is $\Theta(n^{-1/2})$. This result holds under a worst-case skill parameter value model as opposed to the worst-case prior distribution model of this paper. Secondly, both Theorems 1 and 2 hold verbatim if $\mathcal{P}$ is replaced by any set of probability measures with support in $[\delta, 1]$ that contains $\mathsf{unif}([\delta, 1])$.

**Tight minimax bound on skill PDF $P_\alpha$ estimation.** We now state our main result concerning the estimation error for $P_\alpha$. In particular, we argue that the MSE risk of our estimation algorithm (see (7)) scales as $\tilde{O}(n^{-\eta/(\eta+1)})$ for any $P_\alpha \in \mathcal{P}$.

**Theorem 3** (MSE Upper Bound). *Fix any sufficiently large constants $c_2, c_3 > 0$ and suppose that $p \geq c_2 \log(n)/(\delta^5 n)$, $b \geq c_3 \sqrt{\log(n)/n}$, $\epsilon \geq 5 \log(n)/(bn)$, and $\lim_{n\to\infty} \delta^{-1}(npk)^{-1/2} \log(n)^{1/2} = 0$. Then, for any $L_2$-Lipschitz continuous kernel $K : [-1, 1] \to \mathbb{R}$ of order $\lceil \eta \rceil - 1$, there exists a sufficiently large constant $c_{12} > 0$ (that depends on $\gamma$, $\eta$, $B$, $L_1$, $L_2$, and $K$) such that for all sufficiently large $n \in \mathbb{N}$:*

$$R_{\mathsf{MSE}}(n) \leq \sup_{P_\alpha \in \mathcal{P}} \mathbb{E}\left[\int_{\mathbb{R}} \left(\widehat{\mathcal{P}}^*(x) - P_\alpha(x)\right)^2 \mathrm{d}x\right] \leq c_{12} \max\left\{\left(\frac{1}{\delta^2 pk}\right)^{\frac{\eta}{\eta+1}}, 1\right\}\left(\frac{\log(n)}{n}\right)^{\frac{\eta}{\eta+1}}\,.$$

Theorem 3 is established in [34, Appendix B.2]. We next make several pertinent remarks. Firstly, the condition $p \geq c_2 \log(n)/(\delta^5 n)$ is precisely the critical scaling that ensures that $\mathcal{G}(n, p)$ is connected almost surely, cf. [45, Theorem 8.11], [46, Section 7.1]. This is essential to estimate $\alpha_1, \ldots, \alpha_n$ in Step 1 of Algorithm 1, since we cannot reasonably compare the skill levels of disconnected players. Secondly, while $\widehat{\mathcal{P}}^*$ can be negative, the non-negative truncated estimator $\widehat{\mathcal{P}}^+(x) = \max\{\widehat{\mathcal{P}}^*(x), 0\}$ achieves smaller MSE risk than $\widehat{\mathcal{P}}^*$, cf. [5, p.10]. So it is easy to construct good non-negative

estimators. Thirdly, there exists a constant $c_{13} > 0$ (depending on $\eta, L_1$) such that for all sufficiently large $n \in \mathbb{N}$, the following minimax lower bound holds, cf. [20, Theorem 6], [5, Exercise 2.10]:

$$R_{\mathsf{MSE}}(n) \geq \inf_{\hat{P}_{\alpha^n}(\cdot)} \sup_{P_\alpha \in \mathcal{P}} \mathbb{E}\left[\int_{\mathbb{R}} \left(\hat{P}_{\alpha^n}(x) - P_\alpha(x)\right)^2 \mathrm{d}x\right] \geq c_{13}\left(\frac{1}{n}\right)^{\frac{2\eta}{2\eta+1}} \tag{10}$$

where the infimum is over all estimators $\hat{P}_{\alpha^n} : \mathbb{R} \to \mathbb{R}$ of $P_\alpha$ based on $\alpha_1, \ldots, \alpha_n$, and the first inequality holds because the infimum in (3) is over a subset of the class of estimators used in the infimum in (10); indeed, given $\alpha_1, \ldots, \alpha_n$, one can simulate $Z$ via (1) and estimate $P_\alpha$ from $Z$. Thus, when $\eta = 1$, Theorem 3 and (10) show that $R_{\mathsf{MSE}}(n) = \tilde{O}(n^{-1/2})$ and $R_{\mathsf{MSE}}(n) = \Omega(n^{-2/3})$. Likewise, when ($\eta \to \infty$ and) $P_\alpha$ is smooth, i.e., *infinitely differentiable* with all derivatives bounded by $L_1$, Theorem 3 holds for all $\eta > 0$, and an $\Omega(n^{-1})$ lower bound analogous to (10) holds [9]. Letting $\varepsilon = (\eta + 1)^{-1}$, these results yield the first row of Table 1. Fourthly, we note that similar analyses to Theorem 3 can be carried out for, e.g., *Nikol'ski* and *Sobolev classes* of PDFs, cf. [5, Section 1.2.3]. Lastly, it is worth mentioning that BTL models can also be parametrized using *logit parameters* $\omega_i = \log(\alpha_i)$, $i \in [n]$, which are drawn independently from the PDF $P_\omega(x) = e^x P_\alpha(e^x)$, $x \in \mathbb{R}$. When $\delta$ is $\Theta(1)$, it can be shown that the MSE of the estimator $\widehat{\mathcal{P}}^*_{\mathsf{logit}}(x) = e^x \widehat{\mathcal{P}}^*(e^x)$, $x \in \mathbb{R}$ for $P_\omega$ is upper bounded by Theorem 3. Therefore, our analysis of Theorem 3 also holds for estimating distributions of logit parameters.

We emphasize that the key technical step in the proof of Theorem 3 is the ensuing intermediate result.

**Proposition 2** (MSE Decomposition). *Fix any sufficiently large constants $c_2, c_3, c_8, c_9 > 0$ and suppose that $p \geq c_2 \log(n)/(\delta^5 n)$, $b \geq c_3\sqrt{\log(n)/n}$, $\epsilon \geq 5\log(n)/(bn)$, and $\lim_{n\to\infty} \delta^{-1} (npk)^{-1/2}\log(n)^{1/2} = 0$. Then, for any $P_\alpha \in \mathcal{P}$, any $L_2$-Lipschitz continuous kernel $K : [-1, 1] \to \mathbb{R}$, any bandwidth $h \in (0, 1]$ with $h = \Omega\left(\max\{1/(\delta\sqrt{pk}), 1\}\sqrt{\log(n)/n}\right)$, and any sufficiently large $n \in \mathbb{N}$:*

$$\mathbb{E}\left[\int_{\mathbb{R}}\left(\widehat{\mathcal{P}}^*(x) - P_\alpha(x)\right)^2 \mathrm{d}x\right] \leq 2\mathbb{E}\left[\int_{\mathbb{R}}\left(\hat{P}^*_{\alpha^n}(x) - P_\alpha(x)\right)^2 \mathrm{d}x\right] + \frac{c_8 B^2 L_2^2}{h^2}\mathbb{E}\left[\max_{i \in [n]}|\hat{\alpha}_i - \alpha_i|^2\right] + \frac{c_9 L_2^2}{n^5 h^4}$$

*where $\hat{P}^*_{\alpha^n} : \mathbb{R} \to \mathbb{R}$ denotes the classical PR kernel density estimator of $P_\alpha$ based on the true samples $\alpha_1, \ldots, \alpha_n$ (if they were made available by an oracle) [13, 14]:*

$$\forall x \in \mathbb{R}, \quad \hat{P}^*_{\alpha^n}(x) \triangleq \frac{1}{nh}\sum_{i=1}^{n} K\left(\frac{\alpha_i - x}{h}\right). \tag{11}$$

The proof of Proposition 2 can be found in [34, Appendix A.3]. This result decomposes the MSE between $\widehat{\mathcal{P}}^*$ (with general $h$) and $P_\alpha$ into two dominant terms: the MSE of estimating $P_\alpha$ using (11), which can be analyzed using a standard bias-variance tradeoff [5, 20] (see [34, Lemma 4]), and the squared $\ell^\infty$-risk of estimating $\alpha_1, \ldots, \alpha_n$ using (6). To analyze the second term, we use a relative $\ell^\infty$-norm bound from [10, Theorem 3.1] (see [34, Lemma 3]); the same bound was also used to obtain the upper bound in Theorem 1.

## 4 Experiments

We apply our method to several real-world datasets to exhibit its utility. Specifically, Algorithm 1 produces estimates of skill distributions. In order to compare skill distributions across different scenarios as well as capture their essence, it is desirable to compute a single *score* that holistically measures the variation of levels of skill in a tournament.

**Skill score of $P_\alpha$.** Intuitively, a delta measure (i.e., all skills are equal) represents a setting where all game outcomes are completely random; there is no role of skill. On the other hand, the uniform PDF $\mathsf{unif}([0, 1])$ (assuming $\delta$ is very small) typifies a setting of maximal skill since players are endowed with the broadest variety of skill parameters. We refer readers to [16] for a related discussion. Propelled by this intuition, any distance between $P_\alpha$ and $\mathsf{unif}([0, 1])$ serves as a valid score that is larger when luck plays a greater role in determining the outcomes of games. Therefore, we propose to use the negative *differential entropy* of $P_\alpha$ as a score to measure skill in a tournament [42, 47]:

$$-h(P_\alpha) \triangleq \int_{\mathbb{R}} P_\alpha(t)\log(P_\alpha(t))\,\mathrm{d}t = D(P_\alpha\|\mathsf{unif}([0, 1])). \tag{12}$$

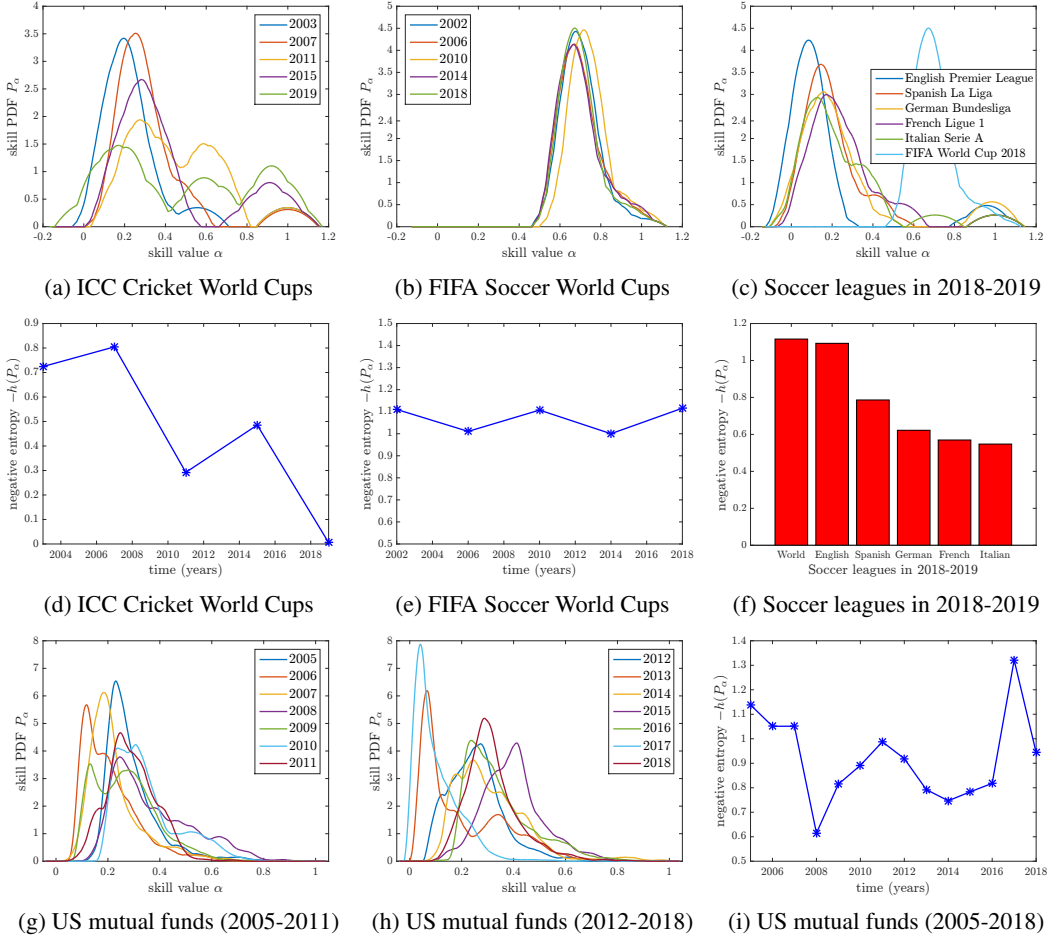

(a) ICC Cricket World Cups  (b) FIFA Soccer World Cups  (c) Soccer leagues in 2018-2019

(d) ICC Cricket World Cups  (e) FIFA Soccer World Cups  (f) Soccer leagues in 2018-2019

(g) US mutual funds (2005-2011)  (h) US mutual funds (2012-2018)  (i) US mutual funds (2005-2018)

Figure 1: Plots 1a, 1b, 1c, 1g, and 1h illustrate the *estimated PDFs* of skill levels of cricket world cups, soccer world cups, European soccer leagues, and US mutual funds, respectively. Plots 1d, 1e, 1f, and 1i illustrate the corresponding *estimated negative differential entropies* of these PDFs.

This is a well-defined and finite quantity that is equal to the *Kullback-Leibler (KL) divergence* between $P_\alpha$ and $\mathrm{unif}([0,1])$ (see [42, Definition 1.4] for a formal definition). To estimate $-h(P_\alpha)$ from data, we will use the simple *resubstitution estimator* based on $\widehat{\mathcal{P}}^*$ and $\hat{\alpha}_1, \ldots, \hat{\alpha}_n$ [15, 48]. (Note that other more sophisticated entropy estimation methods demonstrate the same trends in the sequel.)

**Algorithmic choices.** In all our simulations, we assume that $\eta = 1$, use the Epanechnikov kernel $K_{\mathsf{E}}$, and set the bandwidth to $h = 0.3 n^{-1/4}$; indeed, $h$ is typically chosen using ad hoc data-driven techniques in practice [5, Section 1.4].

**Data processing.** The data is available in the form of wins, losses, and draws in tournaments. For simplicity, we ignore draws and only utilize wins and losses. To allow for 'regularization' in the small data regime, we apply Laplace smoothing so that between any pair of players, each observed game is counted as 20 games, and 1 additional win is added for each player; this effectively means that $p = 1$.

We remark that the constant $0.3$ to define $h$ and the level of smoothing mentioned above are chosen to generate 'smooth' PDFs in Figure 1. Moreover, the qualitative results and trends in the sequel remain the same for a range of values around the constant $0.3$ and the chosen level of smoothing.

**Cricket world cups.** We utilize publicly available data from Wikipedia for international (ICC) Cricket World Cups held in 2003, 2007, 2011, 2015, and 2019. Each world cup has between $n = 10$ to $n = 16$ teams, with each pair of teams playing 0, 1, or (rarely) 2 matches against one another. We learn the skill distributions for each world cup separately as portrayed in Figure 1a. The corresponding negative entropies are reported in Figure 1d. As can be seen, there is a clear decrease in negative

entropy reaching close to $0$ in 2019. This elegantly quantifies sports intuition about the 2019 World Cup having some of the most thrilling matches in the modern history of cricket [3, 4].

**Soccer world cups.** Again, we use publicly available data from Wikipedia for FIFA Soccer World Cups in 2002, 2006, 2010, 2014, and 2018. Each world cup has $n = 32$ teams, with each pair of teams playing 0, 1, or (rarely) 2 matches. Figures 1b and 1e depict the skill distributions and associated negative entropies of soccer world cups over the years. It is evident that the negative entropies have remained roughly constant and away from $0$. This suggests that game outcomes in world cups have remained unpredictable over the years—very consistent with soccer fan experience.

**European soccer leagues.** Yet again, we use publicly available data from Wikipedia for the English Premier League (EPL), Spanish La Liga, German Bundesliga, French Ligue 1, and Italian Serie A in the 2018-2019 season. Each league has between $n = 18$ to $n = 20$ teams, with every pair of teams playing 0, 1, or 2 times against each other (excluding ties). Figure 1c illustrates the skill PDFs of these leagues and the 2018 FIFA World Cup. As expected, we observe that the skill levels of world cup teams are concentrated in a smaller interval closer to $1$. Figure 1f sorts the negative entropies of the skill PDFs and recovers an intuitively sound ranking of these leagues. Indeed, many fans believe that EPL has better "quality" teams than other leagues [1, 2], and this observation is confirmed by Figure 1c. Figure 1c reveals that EPL has higher negative entropy than other leagues since its skill PDF has the tallest and narrowest peak, presumably because EPL only contains high quality teams with little variation among them. This example shows how our algorithm can be used to compare different leagues within the same sport (or even different sports).

**US mutual funds.** Our final experiments are calculated based on data obtained through [49] from *CRSP US Survivor-Bias-Free Mutual Funds Database* that is made available by the Center for Research in Security Prices (CRSP), The University of Chicago Booth School of Business. We consider $n = 3260$ mutual funds in this dataset that have monthly net asset values recorded from January 2005 to December 2018. These values are pre-processed by computing monthly returns (i.e., change in net asset value normalized by the previous month's value) for all funds, which provide a fair measure of monthly performance. Then, we perceive each year as a tournament where each fund plays $k = 12$ monthly games against every other fund, and one fund beats another in a month if it has a larger monthly return. Figures 1g and 1h depict the skill PDFs obtained by applying our algorithm to the win-loss data (produced by the method above) every year, and Figure 1i presents the associated negative entropies. Clearly, 2017 and the Great Recession in 2008 were the times where negative entropy was maximized and minimized, respectively, in Figure 1i. Figures 1g and 1h unveil that the skill PDF is much more spread out in 2008 compared to 2017, which contains a large peak near $0$. So, as expected, far fewer lowly skilled funds existed during the economic recession in 2008. These observations elucidate the utility of our algorithm in identifying and explaining trends in other kinds of data, such as financial data.

## 5   Conclusion

In this paper, we proposed an efficient and minimax near-optimal algorithm to learn skill distributions from win-loss data of tournaments. Then, using negative entropy of a learnt distribution as a skill score, we demonstrated the utility of our algorithm in rigorously discerning trends in sports and other data. In closing, we suggest that a worthwhile future direction would be to develop minimax optimal algorithms that directly estimate entropy, or other meaningful skill scores, from tournament data.

## Broader Impact

The analysis of our algorithm, which forms the main contribution of this work, is theoretical in nature, and therefore, does not have any foreseeable societal consequences. On the other hand, applications of our algorithm to real-world settings could have potential societal impacts. As outlined at the outset of this paper, our algorithm provides a data-driven approach to address questions about perceived qualities of sporting events or other competitive enterprises, e.g., financial markets. Hence, a potential positive impact of our work is that subjective beliefs of stakeholders regarding the distributions of relative skills in competitive events can be moderated by a rigorous statistical method. In particular, our method could assist sports teams, sports tournament organizers, or financial firms to corroborate existing trends in the skill levels of players, debunk erroneous myths, or even unveil

entirely new trends based on available data. However, our work may also have negative consequences if utilized without paying heed to its limitations. Recall that Step 1 of Algorithm 1 estimates BTL skill parameters of agents that participate in a tournament. Since the BTL model is a well-known approach for ranking agents [6,7], it should be used with caution, as with any method that discriminates among agents. Indeed, the BTL model only takes into account wins or losses of pairwise games between agents, but does not consider the broader circumstances surrounding these outcomes. For example, in the context of soccer, the BTL model does not consider the goal difference in a game to gauge how significant a win really is, or take into account the injuries sustained by players. Yet, rankings of teams or players may be used by team managements to make important decisions such as assigning remunerations. Thus, users of algorithms such as ours must refrain from solely using rankings or skill distributions to make decisions that may adversely affect individuals. Furthermore, on the modeling front, it is worth mentioning that the BTL model for pairwise comparisons may be too simplistic in certain real-world scenarios. In such cases, there are several other models of pairwise comparisons within the literature that may be more suitable, e.g., the Thurstonian model, cf. [21], or more general stochastically transitive models, cf. [31]. We leave the analysis of estimating skill distributions or related notions for such models as future work in the area.

## Acknowledgments and Disclosure of Funding

This research was supported in part by the Vannevar Bush Fellowship, in part by the ARO Grant W911NF-18-S-0001, in part by the NSF CIMS 1634259, in part by the NSF CNS 1523546, and in part by the MIT-KACST project.

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
