[Reviews · NeurIPS 2020]

Review 1

Summary and Contributions: - Theoretical: main contribution. The authors show that it is possible, with the proposed algorithm, to estimate a density of a latent variable (skill) from pairwise comparisons whose discrete (win/loss) outcomes are governed by a BTL probability model whose parameters correspond to the latent variables, matching in a nearly-optimal way (up to poly-logarithmic terms) known minimax MSE lower bounds for standard direct density estimation. Significance: High. Other contributions: new minimax lower bound in terms of l_0 and l_inf losses for the skill parameters, showing that the learning rate of the rank centrality algorithm (proposed in prior work and used as a component in the algorithm proposed here) is near-optimal (up to poly-logarithmic terms). - Algorithmic: The algorithm is key to prove the minimax MSE upper bound for the indirect estimation problem. Essentially, it combines two existing algorithms, namely: rank centrality to estimate skills of a BTL model from pairwise comparisons and Parzen-Rosenblatt kernel density estimation. The algorithm is computationally tractable but requires some parameters that in practice are not known, and thus, requires some ad-hoc steps to use it in practice. Significance: Medium. - Methodological: This paper proposes to use the entropy of the latent skill distribution to quantify the role of chance in tournaments, therefore providing a rigourous statistical approach to this problem of importance for gambling regulations. Significance: Medium. In summary, this is not a paper proposing an algorithm with guarantees but more an algorithm that allows to prove an upper bound on the MSE for the indirect density estimation problem. In addition, it proposes the elegant idea of quantifying the role of chance in games by using the entropy of the estimated density.

Strengths: This is a strong theoretical paper with rigorous proofs and proper positioning with respect to prior work. The theoretical results are novel and very significant since they provide an achievable bound for the indirect density estimation problem nearly-matching the direct estimation learning optimal minimax rate. The approach of using the negative entropy of the skill distribution for quantifying the role of chance in tournaments is novel and elegant, while previous work use ad-hoc measures. I think that the work is highly relevant to the NeurIPS community working on Ranking Learning, because the algorithm and the derived bounds constitute an important reference for future work.

Weaknesses: The limitations of this work are essentially along the empirical evaluation axis. The case where the number of comparisons k is variable would deserve a better treatment since it is usually the case as it is shown in the experiments. In practice, the algorithm can't be applied as proposed because the bandwitdh computation requires unknown parameters and therefore, in the experiments, the authors resort to ad-hoc practices that are not detailed. Moreover, in the experiments, an ad-hoc Laplace smoothing is performed to regularize in the small data regime making. The experimental results are illustrative but it is not clear how these ad-hoc decisions impact the results. --- EDIT after response Thank you for your feedback. Your clarification on how to handle the case of variable number of comparisons should help the reader. Regarding the ad-hoc steps, I did see the precise values (note that for the reproducibility question I answered "Yes"). I was referring to how you got these values which is answered by the second part of your answer.

Correctness: I checked high level arguments of the main text and some of the proofs in the Appendix and I didn't find any errors.

Clarity: This is a very well written paper that I enjoyed reading. The math is very well detailed and explained in the Appendix but also properly summarized in the main text. There is also a very good "Related work" section in the appendix that would deserve to be in the main paper if space constraints allowed it.

Relation to Prior Work: It properly positions it in the context of prior work making easy to understand the relevance of the results.

Reproducibility: Yes

Additional Feedback: The case where the number of comparisons k is variable would deserve a better treatment since it is usually the case as it is shown in the experiments. In particular, I suggest to clarify what kind of alteration of the bandwidth should be done when k is variable (Line 151). In general, it would be helpful to explain how the ad-hoc steps are performed since no code is provided. As a suggestion for future work, a possible way of avoiding the ad-hoc steps would be to replace the PR kernel density estimation by a Bayesian or MDL principled approach like in Rissanen, Jorma, Terry P. Speed, and Bin Yu. "Density estimation by stochastic complexity." IEEE Transactions on Information Theory 38.2 (1992): 315-323. In the Mutual Funds experiment, the performance could be considered as a direct observation of skill. Therefore it would be interesting to compare a direct density estimation vs the one based on the (somewhat artificial) pairwise comparisons. Minor comments: By putting the different plot distributions in the same plot, one could be tempted to compare them directly which does not make sense since the skill estimated from different graphs are not comparable. It would be helpful for the reader to clarify this. Line 296: "other" appears twice


Review 2

Summary and Contributions: This paper addresses the problem of learning the probability density function of Bradley-Terry skill parameters given pairwise comparison outcomes. The main contribution of the paper is an algorithm that estimates the PDF of skill parameters with near-optimal minimax error rate. This rate is shown to be no worse than the rate obtained using the skill parameters themselves (i.e., directly using samples of the PDF). Informally, this results shows that one does not "lose much" by having access to skill parameters only through noisy comparison outcomes.

Strengths: To the best of my knowledge, this is the first paper that addresses the question of estimating the distribution from which Bradley-Terry model parameters are sampled (most prior work focuses on reliably estimating the parameters themselves). It builds on and extends a long line of work published at NeurIPS and other machine learning venues. The results presented in this paper are non-trivial and quite elegant. The main take-away—that, in some sense, the skill disitribution can be estimated as efficiently from comparison outcomes as it can be from samples directly—is surprising, and will certainly be of interest to preference learning researchers & practitioners. In passing, the authors, also prove a few bounds on the error of the Rank Centrality, results which are interesting contributions by themselves.

Weaknesses: My main concern / question is about the interpretation of the resulting skill distributions in Section 4. Bradley-Terry models are typically parametrized in one of two different ways: 1. strictly positive parameters, such that the ratio of the parameters corresponding to two items equals the odds of one item winning over the other (let us call these parameters "skills") 2. real-valued parameters, such that the probability of one item winning over the other is equal to the logistic function of the difference in parameters (let us call these parameters "logits") In this paper, the authors work with the parametrization (1). I would argue that comparing entropies of distributions in that parametrization leads to results that are somewhat difficult to interpret. In particular, it is not clear to me what a uniform skill distribution over [eps, 1] means in terms of expected outcomes. Indeed, the comparison outcome probabilities between teams with skill eps vs. 2*eps are comparable to those between teams with skills 0.5 and 1. But a uniform distribution puts much more mass on the interval (0.5, 1) as it does on the interval (eps, 2*eps). In my opinion, Figures 1c and 1f illustrate this: "World" and "English" seem to have very different skill distributions (in terms of "fan experience"), yet result in a similar differential entropy scores. I suspect it is much easier to interpret distributions over logits. Do the authors have any comments on this? --- EDIT after response Thank you for your feedback. In terms of applications / interpretation, I am still not convinced: if there are n >> 2 teams - a uniform distribution over logits has a fairly intuitive interpretation. In expectation, prob(i wins over j) depends only on the rank difference between teams i and j - Whereas a uniform distribution over skills is hard to make sense of. In expectation, match outcomes will be much "noisier" / uncertain between teams at the top of the ranking, much less so for teams at the bottom of the ranking. As such, I believe the link between skill distribution entropy and sports intuition & fan experience could still be clarified. Nevertheless, since the rest of the results developed in the paper are rigorous, strong & interesting, I continue to be very much in favor of accepting this paper.

Correctness: I did not check the proofs systematically. The main arguments behind the various results appear to be plausible.

Clarity: The paper is very well written and was a pleasure to read.

Relation to Prior Work: Even though the related work section is deferred to the supplementary material, the authors do a good job at positioning their work in the context of related literature "along the way", as they introduce the problem and develop their method.

Reproducibility: Yes

Additional Feedback:


Review 3

Summary and Contributions: This paper regards learning the skill distribution of a population of agents from pairwise games under the assumption that outcome likelihoods are governed by a multinomial logit model. At a high level, the paper’s algorithm first estimates the skill parameters of each agent, then uses these estimate parameters to estimate the distribution from which they were sampled. The paper suggests that the entropy of the distribution provides a quantitative measure of skill/competitiveness/excitingness. To support this claim, the paper analyzes data from cricket, soccer, and mutual funds.

Strengths: Overall, the reviewer feels that the paper is well-written and does a good job at explaining the important ideas in language understandable to those without a statistical background. The paper’s goals, claimed contributions, and experiments are all clear.

Weaknesses: The reviewer has minor criticisms regarding the paper’s use of the language skill, excitingness, and competitiveness. Calling negative the negative entropy of a skill distribution the “overall skill score” reads as if it is a measurement of the skill of the participants, whereas it is meant to be read as a measurement of the variation of the skill of the participants. Suggesting that this corresponds to excitingness also seems not quite right. Many pundits feel that recent NBA seasons have been exciting, despite that there were massive skill discrepancies between the top few teams and the rest of the league. It may also be the case that a league of closely matched teams has many one-sided/non-competitive games, or that a league of not closely matched teams has many close games.

Correctness: The reviewer does not possess the statistical prowess to follow the paper’s theoretical results.

Clarity: Yes.

Relation to Prior Work: Yes.

Reproducibility: Yes

Additional Feedback: The reviewer has one significant question of the paper regarding the application of its method as a quantitative measurement. Why is the entropy of the estimated skill distribution from which the samples were drawn preferred to the entropy of the estimated empirical distribution? The reviewer’s intuition is that the latter would better quantify the skill/competitiveness/excitingness of the games. If this intuition is correct, what reason is there to estimate to estimate the underlying skill distribution? The reviewer’s opinion of this paper depends on the answer to this question. The reviewer is very willing to raise the assessed score accordingly. AFTER REBUTTAL: The reviewer's main concern with the submission was the motivation behind estimating the skill PDF. Based on the rebuttal and the other reviews, it seems that this motivation is well-founded. The reviewer increased the assessed score accordingly.


Review 4

Summary and Contributions: This paper proposes an algorithm to learn BTL model. Different from the usual setting, this paper assumes that the parameter of each item (team, fund manager, etc.) follows a distribution and the proposed algorithm aims at estimating this distribution. This distribution gives information on how different the items are. The algorithm has two steps: estimating the parameter for each item, and estimating the distribution using the estimated parameter in the first step. The authors provide theoretical bound on the accuracy of the algorithm and some experiments that validate their motivation.

Strengths: The main contribution of this paper is theoretical. Theorems 1 and 2 aims at characterizing the accuracy of learning the parameter of BTL and Theorem 3 characterizes the accuracy of estimating the overall distribution. I did not check the proofs, but these theorems seem nontrivial.

Weaknesses: 1. This work is not well-motivated. It seems estimating the distribution is not necessary since the variance of the learned parameters for each item (teams, fund managers, etc.) already indicates whether the parameters of all items closely center somewhere or spread uniformly. So I don't see the point of a sophisticated second step in the algorithm. This paper does not provide a comparison in the experiments between their approach and the naive approach mentioned above either. 2. Most of this paper seems very technical but the theorems do not qualify as theorems. "sufficiently large constants" and "sufficiently large n" appear in every theorem statement but the authors fail to specify how large is sufficient. These are necessary for theorems.

Correctness: I don't see obvious errors, but the statements of theorems need to be rewritten.

Clarity: Yes.

Relation to Prior Work: Yes.

Reproducibility: Yes

Additional Feedback: I appreciate the authors' response and have updated my score accordingly.

[Author Response · NeurIPS 2020]

Thank you for all your comments. Our responses are detailed below, and we will incorporate them in the final paper.

**Reviewer 1: Variable number of comparisons $k$.** To simplify notation and analysis, it's standard in the BTL literature
to assume that pairs are compared a fixed number $k$ times (see [10] or [12]). As explained in lines 150-152, "if $k$ varies
between players so that $i$ and $j$ play $k_{i,j} = k_{j,i}$ games," the data $Z$ can be normalized accordingly for our algorithm.
The bandwidth $h$ can be altered, e.g., to use $k' = \min_{\{i,j\} \in \mathcal{G}(n,p)} k_{i,j}$ in place of $k$ in equation (8). This would yield
corresponding theoretical guarantees with $k'$. We will clarify this in the final paper. One could also carry out the long
analysis in [10] (used in Lemma 3) and our analysis while keeping track of the $k_{i,j}$'s, and derive a corresponding $h$ that
yields finer bounds. We decided to omit this due to space constraints.

**Reviewer 1: Ad-hoc steps in experiments.** We have already explained all the details needed to run all our experiments
in the paper. Perhaps the reviewer missed these details. Please see lines 150-151 and 238-246 for the precise values and
choices used to execute our algorithm, e.g., $h = 0.3 n^{-1/4}$. On the other hand, we had not mentioned that we chose the
constant $0.3$ in $h$, and the level of smoothing ("... game is counted as 20 games ..."), by eyeballing when the densities
in Figure 1 looked 'smooth.' Moreover, our qualitative results and trends in section 4 remain the same for a range of
values around $0.3$ and $20$ (e.g., $0.4$ or $30$). We will clarify these points in the final paper.

**Reviewer 1: Future directions and minor comments.** The minor corrections and clarifications will be made in the
final paper. Thank you very much for suggesting future research avenues pertaining to Bayesian or MDL approaches
and density estimation based on mutual fund performances.

**Reviewer 2: Interpreting skill PDFs.** We assume a lower bound on our skill PDFs over $[\delta, 1]$ in a neighborhood of $1$
(see line 114). This implies that $\max_i \alpha_i \approx 1$ with high probability for large $n$ (see (31) in supplementary materials).
Intuitively, we are re-normalizing all skill parameters so that the maximum one is essentially $1$. So, if there are just two
teams with skills $\delta$ and $2\delta$, these values will be re-normalized to $0.5$ and $1$. Since $\max_i \alpha_i \approx 1$, it is reasonable for the
uniform skill PDF to put more mass on larger intervals, i.e., the uniform skill PDF is interpretable. Thus, we do not see
any immediate advantage of using logits. In Figures 1c and 1f, "World" and "English" have different skill PDFs but
similar skill scores, because different PDFs can have the same KL divergence to the uniform PDF. This artifact remains
even if we use logits. On a separate note, since the logits $\omega_i = \log(\alpha_i)$ are i.i.d. with PDF $P_\omega(t) = e^t P_\alpha(e^t)$, when $\delta$
is constant and $h < \delta$, we have an estimator $\widehat{\mathcal{P}}_2^*(t) = e^t \widehat{\mathcal{P}}^*(e^t)$ for $P_\omega$ if desired. By substitution, $\mathbb{E}\big[\int (\widehat{\mathcal{P}}_2^* - P_\omega)^2\big]$
$\leq (1+\delta)\,\mathbb{E}\big[\int (\widehat{\mathcal{P}}^* - P_\alpha)^2\big]$. Therefore, the upper bound in Theorem 3 also holds for MSE estimation of logit PDFs.

**Reviewers 3 & 4: Why not estimate skill score directly from $\hat{\alpha}_1, \ldots, \hat{\alpha}_n$ (instead of estimating skill PDF $P_\alpha$)?**
We outline several reasons to estimate $P_\alpha$: (i) If one seeks to estimate a specific functional of $P_\alpha$, e.g., moments or
variance, it is possible to estimate this directly from $\hat{\alpha}_1, \ldots, \hat{\alpha}_n$. (This is still nontrivial because careful analysis is
needed to prove consistency of estimation based on noisy pairwise comparisons.) However, $P_\alpha$ *contains information*
*about all such functionals* and provides a lot more qualitative information as shown in Figure 1. Since different
functionals are needed for different applications, a good estimate of $P_\alpha$ rather than just samples is very useful. Our
main contribution is showing that the entire smooth density $P_\alpha$ can be estimated from $\hat{\alpha}_1, \ldots, \hat{\alpha}_n$ as well as if we had
access to the true $\alpha_1, \ldots, \alpha_n$. (ii) The dual characterization of TV distance and the Cauchy-Schwarz inequality give:
$T \triangleq \sup_{P_\alpha, \|f\|_\infty \leq 1} \mathbb{E}\big[(\int f \mathrm{d}\widehat{\mathcal{P}}^* - \int f \mathrm{d}P_\alpha)^2\big] \leq \sup_{P_\alpha} \mathbb{E}\big[(\int |\widehat{\mathcal{P}}^* - P_\alpha|)^2\big] \leq 3 \sup_{P_\alpha} \mathbb{E}\big[\int (\widehat{\mathcal{P}}^* - P_\alpha)^2\big]$, where the first
sup is over $P_\alpha$ and all functions $f$ bounded by $1$. Thus, the bound in Theorem 3 holds for $T$. So, by estimating $P_\alpha$, we
obtain *uniform guarantees on estimating any bounded statistic* of the form $\mathbb{E}[f(\alpha)]$, which includes all moments. (iii)
We believe differential entropy $h(P_\alpha)$ is an excellent overall skill score, and standard non-parametric estimators for it in
the literature (e.g., integral, resubstitution, or splitting data estimators) *require an estimate of $P_\alpha$* first to plug in. (Also,
quantization theory shows that discrete "entropy of the empirical distribution" of $\hat{\alpha}_1, \ldots, \hat{\alpha}_n$, with uniform binning, is
a poorer estimate of $h(P_\alpha) - \log(\text{bin size})$.) (iv) Philosophically, we believe that skill levels of players, like height or
weight, exhibit a distribution of values, and a tournament contains samples from this distribution. Hence, the "right"
skill score is based on $P_\alpha$ rather than the realizations $\alpha_1, \ldots, \alpha_n$. We will elaborate on these points in the final paper.

**Reviewer 3: "Skill" vs. "exciting" vs. "competitive".** We will clarify that the differential entropy based "overall skill
score" measures variation of skills, not the actual skills of players, in the final paper. We used "exciting" sparingly in the
paper, but agree that it may not precisely capture what we mean. So, we will change or clarify it in technical discussions
in the final paper. We have also sparingly referred to tournaments with high overall skill scores as "competitive,"
because many teams have similar skill parameters and game outcomes are less predictable. This usage seems reasonable
and we have retained it. We also agree that closely matched teams may have many non-competitive games, and our
overall skill score is indeed an "average measure" which may not capture these low probability events.

**Reviewer 4: Theorem statements.** Using the phrase "sufficiently large" is *standard practice* in mathematical statistics
(see, e.g., [10]), and it has a very *precise meaning*. "Sufficiently large $n$ (or constant $c$)" means that "there exists a
constant $A$ such that for all $n \geq A$ (or $c \geq A$)." Here, the values of $A$ may depend on other constant problem parameters,
and they can be deduced from our proofs. For example, in Theorem 1, the constant $c_{15} \geq 2 c_4/(\delta \sqrt{pk})$, where $c_4$ is the
universal constant from Lemma 3 (which is Theorem 3.1 in [10]). Moreover, for "large constants," we already mention
which parameters $A$ depends on in the paper. Since we do not derive sharp values of $A$, it is not illustrative to include
them in theorem statements. However, our theorem statements are rigorous; e.g., they directly imply big-$O$ style results.

[Meta-Review · NeurIPS 2020]

The entire review team is in agreement that this is a solid paper. It deals with a problem that is motivated, and contains results comprising strong theory and interesting experiments. Note: I have discounted the reviewer comment "Most of this paper seems very technical but the theorems do not qualify as theorems. "sufficiently large constants" and "sufficiently large n" appear in every theorem statement but the authors fail to specify how large is sufficient. These are necessary for theorems.". This is a standard practice in learning theory. We the authors to please incorporate the following in the camera ready: - Please do clarify in the main text on how to handle variable numbers of comparisons - See other excellent suggestions by reviewer 1 - Reviewer 1 brings up the important point that estimation of skills in practice may have more far reaching broader impact, and hence issues pertaining to the assumed models are important. The review team notes that the paper does an excellent job in discussing some of this in the broader impacts section pertaining to issues like injury to etc. Additionally on the technical front, we suggest the revision should recognize the limitation of using the simplistic vanilla BT models instead of models having covariates, or stochastically transitive models, or non-transitive models. These may anyways be exciting to the ML community as possible future work given the recent interest in such models in the community. - Please do clarify the issue pertaining to uniform distribution of skills. Please see "EDIT after response" in review 2. - Move some points about the motivation from the rebuttal to the main text in order to avoid concerns like what reviewers 3 and 4 brought up in their initial reviews.